# Treatment with Myo-Inositol Does Not Improve the Clinical Features in All PCOS Phenotypes

**DOI:** 10.3390/biomedicines11061759

**Published:** 2023-06-19

**Authors:** Vittorio Unfer, Michele Russo, Cesare Aragona, Gabriele Bilotta, Mario Montanino Oliva, Mariano Bizzarri

**Affiliations:** 1UniCamillus-Saint Camillus International University of Health Sciences, 00131 Rome, Italy; 2The Experts Group on Inositol in Basic and Clinical Research (EGOI), 00161 Rome, Italy; aragonacesare@gmail.com (C.A.); dr.montanino@gmail.com (M.M.O.); mariano.bizzarri@uniromai1.it (M.B.); 3R&D Department, Lo.Li. Pharma, 00156 Rome, Italy; m.russo@lolipharma.it; 4Systems Biology Group Lab, 00161 Rome, Italy; 5Alma Res Fertility Center, 00198 Rome, Italy; gabriele.bilotta90@gmail.com; 6Department of Obstetrics and Gynecology, Santo Spirito Hospital, 00193 Rome, Italy; 7Department of Experimental Medicine, University La Sapienza, Via A. Scarpa 16, 00160 Rome, Italy

**Keywords:** PCOS, Rotterdam criteria, myo-Inositol, PCOS phenotypes, metabolic syndrome

## Abstract

Background: The aim of the present study is to investigate the effects produced by a treatment with myo-Inositol (myo-Ins) in women presenting polycystic ovary syndrome (PCOS) of different phenotypes. Methods: We performed a retrospective study to evaluate whether patients presenting different PCOS phenotypes, treated for 6 months with myo-Ins, might exhibit a differential response to the treatment. On this premise, we clustered women with PCOS phenotypes A, B, and C in the first study group (hyperandrogenic PCOS or H-PCOS), and women presenting PCOS phenotype D in a separate study group (non-hyperandrogenic PCOS or NH-PCOS) to evaluate if the presence of hyperandrogenism, shared by H-PCOS, might imply a metabolic/endocrine condition rather than a gynecological issue. Results: The administration of myo-Ins induced a significant improvement in metabolic and endocrine parameters in H-PCOS, while the effects on NH-PCOS were negligible. Additionally, myo-Ins treatment improved the endometrial thickness of H-PCOS. Conclusions: Subjects selected for the study exhibited a differential response to myo-Ins therapy according to their PCOS phenotypes. The data suggest that the same treatment might not equally improve the parameters of the PCOS condition in each sub-group of patients. It is crucial to distinguish the various phenotypes to properly select the therapeutical approach.

## 1. Introduction

Polycystic ovary syndrome (PCOS) is one of the most diffused pathological conditions in women of reproductive age, showing a worldwide prevalence of 5–15%, and is characterized by a series of endocrinologic and metabolic alterations [1]. Patients affected by PCOS are usually diagnosed according to the Rotterdam criteria, which outlines four possible phenotypes. Indeed, PCOS is diagnosed when at least two out of three of the following features are detected: oligo- or anovulation, clinical and/or biochemical hyperandrogenism, or polycystic ovaries on ultrasound analysis (Figure 1) [2]. Women affected by this condition often exhibit esthetic manifestation of hyperandrogenism such as acne, hirsutism, hypertrichosis, seborrhea, and alopecia. They also present metabolic dysfunction associated with obesity, insulin resistance, hyperinsulinemia, and diabetes in addition to various reproductive issues, and psychological disorders [3]. Rotterdam classification specifies differences in the various phenotypes, which translate into distinct manifestations of the condition. However, even if women are classified with different phenotypes of PCOS, it is quite common that these individuals are considered a homogenous group for both clinical studies and for treatment recommendations [4]. Considering the scientific literature already published, it should be considered that the individuals enrolled in clinical trials are not differentiated into separate groups according to their phenotypes. This means that different phenotypes are present together in the study groups; for instance, hyperandrogenic women are pooled together with non-hyperandrogenic women, or women with a polycystic ovarian morphology (PCOM) with non-cystic ovarian morphology. The heterogeneity of the subjects included in clinical trials might affect their findings and may also cast doubt on the real efficacy of the treatment investigated. 

In recent years, a plethora of data have supported the effectiveness of the administration of insulin sensitizers for the management of PCOS symptoms, such as myo-inositol (myo-Ins) and metformin [5,6]. In particular, myo-Ins is a natural molecule present in fruit and vegetables [7] and is a fundamental player in the physiology of the female reproductive apparatus [8] and pregnancy [9]. The supplementation of myo-Ins in PCOS patients undergoing assisted reproductive technique (ART) programs improves oocyte development and maturation [10]. Specifically, treatment with myo-Ins is recognized for the improvement in the hyperandrogenic profile of women with PCOS and the resulting esthetic manifestations. Hence, myo-Ins and its stereoisomer D-chiro-inositol (D-chiro-Ins), are actively involved in the follicular development process. [11]. When correct myo-Ins/D-chiro-Ins ratios are preserved in the follicles, the physiology of the folliculogenesis is maintained. In contrast, an altered inositol’s ratio may affect the ovulation process, reduce the oocyte quality, inhibit correct signaling of the follicle-stimulating hormone (FSH), and lead to androgen overproduction [12]. Additionally, impairment of the physiological oocyte maturation can lead to the formation of typical ovarian cysts, a fundamental marker of the syndrome [13].

In overweight or obese women, treatment with both stereoisomers, which reflect the physiological ratio of myo-Ins/D-chiro-Ins (40:1) detected in the plasma, seems to be effective in rebalancing the cellular content of myo-Ins in the ovary while maintaining the correct ratio with D-chiro-Ins [14]. Moreover, myo-Ins administration in women with PCOS ameliorates both hormonal and metabolic parameters [15] by transducing insulin signaling and lowering insulin plasma levels [16].

The positive effect of myo-Ins treatment for PCOS is supported by a great number of published works [17,18,19,20]; however, myo-Ins therapy is not currently tailored to the different PCOS phenotypes, failing to consider their specific characteristics [21]. In particular, women presenting PCOS with phenotype D do not exhibit hyperandrogenism, unlike those with PCOS with phenotypes A, B, and C, thus indicating that differences in phenotype may reflect differences in the etiopathogenesis of the syndrome. In this regard, the present investigation compares the effects of myo-Ins treatment in different PCOS phenotypes, in order to evaluate the specificity of the treatment and investigate the effects of this approach on different types of patients.

## 2. Materials and Methods

A non-randomized uncontrolled retrospective observational study was conducted, intending to test the same treatment of myo-Ins in two groups of patients with PCOS with different features. The scope of this study was to test whether the different patients selected for the study exhibit different responses to the same treatment of myo-Ins.

Thirty-five women were selected for the study according to following inclusion criteria: diagnosis of PCOS based on the Rotterdam criteria; age 18–40; BMI 20–35; therapy with myo-Ins at least for 6 months. Exclusion criteria were as follows: presence of co-morbidities, pregnancy, previous treatments with inositols or insulin sensitizers and/or hormonal, BMI > 35, history of alcohol or drug use.

Since the present investigation is not based on previous studies regarding the use of myo-inositol in different PCOS phenotypes, we did not calculate sample size. The patients selected for the present study were chosen from our database according to the inclusion/exclusion criteria described above and received myo-Ins treatment for at least 6 months between March 2021 and January 2023.

Selected subjects were divided in two different groups according to the diagnosis of PCOS and phenotype identification.

The first group (16 patients) included women with phenotypes A, B, and C and therefore was indicated as hyperandrogenic PCOS (H-PCOS), while the second group (19 patients) included only women with phenotype D and thus was indicated as non-hyperandrogenic PCOS (NH-PCOS). Both groups were supplemented with a supplement of: myo-inositol 2000 mg, folic acid 200 µg, α-lactalbumin 50 mg, twice daily for 6 months.

The measured parameters were as follows: body mass index (BMI), homeostasis model assessment for insulin resistance (HOMA), level of insulin, glucose or glycemia, follicle-stimulating hormone (FSH), luteinizing hormone (LH), testosterone, sex-hormone-binding globulin (SHBG), cholesterol, and triglycerides.

The values of the analyzed parameters were recorded at baseline (T0) (results showed in Table 1) during the first visit, and after six months of treatment period (T6) with myo-Inositol.

Ultrasonographic measurement of endometrial thickness was carried out between day 3 and 6 of the menstrual cycle.

The Internal Review Board of the ethical committee of the Alma Res Clinic approved the study. The study is registered with code: NCT05678114 on clinicaltrials.gov.

The statistical analysis was performed with a Wilcoxon signed rank test to evaluate the treatment efficacy in each group over the treatment period, and a Mann–Whitney U test was used to compare variations between the two groups considered in the study. The results are reported as median values with the related 25th and 75th percentiles.

## 3. Results

### 3.1. Patients with PCOS Phenotype A-B-C (H-PCOS)

The treatment with myo-Ins significantly improved glycemia and the HOMA index after 6 months, highlighting myo-Ins’s ability to rebalance glucose homeostasis and ameliorate insulin signaling (Table 2). The treatment also induced a decrease in BMI and a slight, but non-significant, increase in insulin levels. Additionally, a significant reduction was observed in the levels of total testosterone, in addition to a significant increase observed in SHBG levels. Hormonal parameters are strongly dependent upon myo-Ins levels, particularly in H-PCOS patients. A significant reduction was also reported in the LH/FSH ratio and in triglycerides.

The median value of the endometrial thickness measured at baseline was 3 mm; however, it showed a significant increase after 6 months of myo-Ins treatment, reaching a median value of 6 mm. As such, myo-Ins treatment appears to be effective in mediating estrogen-dependent endometrial proliferation and thickening.

### 3.2. Patients with PCOS Phenotype D (NH-PCOS)

After 6 months of myo-Ins supplementation, the HOMA index and the insulin levels significantly diminished, which further indicates the role of myo-Ins in the rebalancing of glucose homeostasis and in ameliorating insulin signaling. Over the treatment period, myo-Ins induced a slight, but not significant, decrease in BMI and glycemia (Table 2).

Moreover, a significant increase was observed in SHBG with no significant variation in testosterone levels. The variation induced in testosterone was not significant, and baseline values were notably lower in this study group with respect to H-PCOS patients. A significant variation was described in SHBG levels. However, myo-Ins induced minor variations in the sex hormones in NH-PCOS with respect to the group of H-PCOS.

After 6 months of treatment, no significant changes were seen in the levels of LH, FSH, and the LH/FSH ratio, thus indicating a minor effect of myo-Ins on the gonadotrophins in NH-PCOS (Table 2). Moreover, no significant change in endometrial thickness was observed following a 6-month treatment of myo-Ins.

### 3.3. H-PCOS Patients vs. NH-PCOS Patients

A significant difference between the two groups was observed in glucose levels and the HOMA index, thus suggesting that myo-Ins was more effective in the H-PCOS group (Table 3). Additionally, the same trend was observed in terms of testosterone and SHBG variation, which were significantly higher in H-PCOS.

The LH/FSH ratio and the levels of triglycerides were also more significantly altered in the H-PCOS group in comparison to the NH-PCOS patients.

The measured endometrial thickness was significantly different between the two treatment groups following myo-ins supplementation. In the H-PCOS group, the observed median change was 3, compared to −1 in the NH-PCOS group. Taken together, the results confirm a stronger activity of myo-Ins treatment in H-PCOS patients.

## 4. Discussion

The present study demonstrates that different phenotypes of PCOS might represent distinct clinical conditions rarely separated in the published literature. Even if the sample size is relatively small, the subjects selected for the study exhibited interesting differences in the evaluated parameters, which is reflected in a differential response to myo-Ins treatment. Indeed, after 6 months of myo-Ins administration, an improvement in hormonal and metabolic parameters was observed in H-PCOS, while a negligible effect was described in NH-PCOS. These findings are in line with the literature describing that myo-Ins levels are fundamental for the balancing of the endocrine/metabolic profile in PCOS patients [15]. The effects mostly observed in H-PCOS possibly indicate that this group exhibits higher dysmetabolic and impaired endocrine status compared to NH-PCOS. In addition, evidence from the literature indicates a strong correlation between myo-Ins and hyperandrogenism regarding PCOS. Indeed, myo-Ins administration significantly improved androgen parameters and significantly diminished the esthetic manifestations such as acne and hirsutism [22,23]. In the present study, we observed that myo-Ins supplementation was strongly associated with a reduction in testosterone levels and an increase in SHBG in H-PCOS. This provides further support to the idea that myo-Ins may have a positive role in the improvement in the hormonal parameters related to hyperandrogenism in PCOS patients, as previously mentioned in the literature [24,25].

Conversely, a reduced effect was observed in NH-PCOS since these women are not characterized by hyperandrogenism and seem less dysmetabolic compared to H-PCOS.

The fundamental role of myo-Ins in PCOS has been endorsed worldwide for more than 20 years [26], but this is the first retrospective study comparing the effects of myo-Ins treatment in different PCOS phenotypes.

The present investigation is an extension of a previous publication from our research group, where we suggested a new perspective of both etiopathogenesis and the clinical evaluation of PCOS [27]. Specifically, we questioned the term PCOS and its meaning, highlighting that differences observed in the four phenotypes of PCOS may reflect a different onset of the pathology.

Data retrieved from the scientific literature indicate that about 75% of women with PCOS exhibit insulin resistance (IR) [28], or other endocrine and metabolic alterations such as hyperinsulinemia, diabetes, and obesity [29,30]. Even if there is very limited literature properly addressing the differences between the various phenotypes, preliminary indicators suggest that women presenting PCOS with phenotype D (NH-PCOS) seem more likely to be normo-metabolic compared to the other phenotypes, thus indicating that differences between individuals with different PCOS phenotypes exist [31]. With this premise, we investigated in the present study if the differences between the phenotypes described in the literature are also detected in the subjects selected for the study.

The values recorded in our study at baseline confirmed differences between the two study groups, suggesting that H-PCOS exhibit enhanced metabolic and endocrine imbalance compared to NH-PCOS. Specifically, we observed a significant difference in BMI, insulin, and glucose levels, with higher values observed in H-PCOS phenotypes. The evaluation of IR also revealed a significantly higher starting HOMA index in the H-PCOS group compared to the NH-PCOS cohort. Moreover, the baseline value of the LH/FSH ratio was significantly higher in the H-PCOS group, confirming an existing imbalance of hormonal and metabolic factors.

Despite the relatively small population evaluated for the present study and the preliminary nature of this work, our data are in line with the findings published by Moghetti et al. [22], describing lower metabolic and hormonal alterations in NH-PCOS compared to the other phenotypes.

A fundamental aspect regarding the PCOS phenotypes is that hyperandrogenism is a shared feature of phenotypes A, B, and C but not phenotype D (Figure 1) [32]. Considering this, we clustered phenotypes A, B, and C as H-PCOS, separating the PCOS phenotype D patients into a separate group, namely NH-PCOS. The measurement of testosterone levels at baseline confirms a significant difference between the two study groups, with a higher value detected in H-PCOS patients. Androgen excess is strongly correlated with irregularity of the menstrual cycle and often coexists with an altered ovarian physiology, which in turn may hinder the occurrence of a physiological pregnancy [33]. Since a high percentage of women presenting PCOS also exhibit metabolic issues such as IR, hyperinsulinemia, obesity, and diabetes, it may be hypothesized that hyperandrogenic manifestations rely on alterations in the endocrine and metabolic pattern [34,35].

Insulin directly stimulates the production of androgens in the ovarian theca cells and additionally inhibits the synthesis of SHBG in the hepatocyte, thus causing an increase in the levels of circulating free androgens [26]. Hence, high insulin levels and IR directly impair ovarian function by inducing an upregulation of androgen production [36].

This hypothesis is coupled with the idea that if hyperandrogenism is absent only in NH-PCOS, it is therefore plausible that the pathology has a different origin. The presence of both PCOM and oligo- or anovulation with very few indications of dysmetabolism in these subjects, could indicate a syndrome of gynecological origin and a real ovarian pathology not involving hormonal or metabolic alterations.

The therapeutic rationale behind the effectiveness of myo-Ins administration in PCOS patients relies on its recognized activity in the ovaries as a mediator of FSH and insulin signaling [37]. A deregulation of ovarian myo-Ins quantities may affect FSH signaling, as frequently described in PCOS patients, and impair the ovulation phase [38]. Lower levels of intraovarian myo-Ins in a PCOS context could also affect insulin signaling and glucose uptake, thus unbalancing the metabolism of both oocytes and follicular cells. Considering that oocytes exhibit high glucose consumption in the maturation process, this may strongly hamper oocyte development and reduce oocyte quality [39].

As an insulin sensitizer, myo-Ins is able to improve insulin signaling of target tissues, increasing glucose uptake in the human body. This activity allows a reduction in insulinemia and hyperinsulinemia frequently observed in PCOS patients, resulting in a positive effect on the reproductive apparatus and hormonal pattern. Additionally, myo-Ins may improve the reproductive functions by mediating the release of the Ca^2+^ ions during oocyte development. This process is necessary for the acquisition of meiotic competence and to sustain the final stages of oocyte maturation [40]. In the present study, treatment with myo-Ins induced a significant improvement in glucose levels and the HOMA index in H-PCOS, but no significant variation in insulin status and BMI. In the NH-PCOS group, a significant variation in insulin and the HOMA index was recorded, but no variation in glucose levels was observed. These data suggest that in some PCOS patients, a therapeutic intervention may be insufficient without proper lifestyle modification [41]. Hence, the diet of patients included in the patient subset was not monitored, particularly as the H-PCOS group had a higher average BMI at baseline than the NH-PCOS cohort. However, we reported a significant improvement in the HOMA index for H-PCOS, and the variation was higher with respect to the variation described in NH-PCOS, which means that insulin sensitivity is strongly ameliorated with myo-Ins treatment.

In line with the current literature, myo-Ins is a beneficial therapy to ameliorate insulin sensitivity, to improve the LH/FSH ratio, and to reduce the biochemical markers of hyperandrogenism [22]. However, these data were collected from studies referring to PCOS subjects not stratified according to their phenotypes, but rather as a single study group. Interestingly, the variations detected for the HOMA index, glycemia, LH/FSH ratio, testosterone, and SHBG in the present study were significantly higher in H-PCOS with respect to NH-PCOS. The results recorded after 6 months of myo-Ins administration clearly indicate a more prominent effect of the treatment in subjects presenting hyperandrogenism.

These findings suggest that it seems reasonable to hypothesize that hyperandrogenism and its dermatological presentations in H-PCOS may derive from endocrinological and metabiological issues rather than a gynecological condition, supported by the H-PCOS group’s response to myo-Ins treatment.

Furthermore, if we exclude a condition of hyperandrogenism in NH-PCOS, we should also evaluate that an alternative mechanism affecting ovarian physiology could exist in these subjects. An interesting finding of the present study was a significant difference observed in the endometrial thickness between the two groups. We highlighted a higher baseline value in NH-PCOS (7.9 mm) compared to H-PCOS (2.9 mm). Considering that the median value of a normal endometrial thickness measured between day 1 and 6 of the menstrual cycle is 7.0 mm [42], our data indicate a slight thickening of the endometrium in NH-PCOS. Excluding hyperandrogenism, it might be hypothesized that those patients could be affected by a form of local or relative hyperestrogenic status, or even by a reduced progesterone activity, which causes the thickening of the endometrium and the anovulation issue.

Additionally, myo-Ins induced a significant increase in the endometrial thickness in H-PCOS and a negligible effect in the group of NH-PCOS. It might be hypothesized that myo-Ins counteracts the effect of the excessive testosterone levels in the ovary, by enhancing the FSH receptor and aromatase synthesis in granulosa cells [43], thus contributing to the rebalancing of the estrogen levels. This mechanism might contribute to the reduction of the impact of hyperandrogenism in these phenotypes and allow the increase in the estrogen-dependent endometrial thickness [44].

On this premise, it should be considered that the phenotypes of PCOS, may likely represent different clinical conditions, so that a revision of the diagnostic criteria seems to be required. Updating of the diagnostic criteria also means revising the therapeutic rationale related to PCOS, which should suggest a tailored approach for the diverse clinical conditions observed in different patients, currently indicated collectively as women with PCOS.

Our study provides an innovative overview of the PCOS condition since no other clinical study has ever tested the same treatment in diverse PCOS phenotypes. The differential response recorded in H-PCOS vs. NH-PCOS is a preliminary indication that may reflect an alternative etiopathogenesis of the syndrome. However, further analysis including double-blinded, randomized, controlled, prospective trials and larger scale investigation is merited. Additionally, our speculation on the endometrial thickness needs to be further clarified, probably associating the measurement of the endometrium with the evaluation of estradiol levels.

## 5. Conclusions

The current understanding of the PCOS condition frequently fails to consider the existing differences between the various phenotypes described by the Rotterdam criteria. We supplemented two study groups with myo-Ins, sorting the patients exhibiting hyperandrogenism (phenotype A-B-C or H-PCOS) from the non-hyperandrogenic patients (phenotype D or NH-PCOS). The treatment investigated in the study, significantly improved the hormonal and endocrine profile of H-PCOS, while it was less effective in NH-PCOS with a diminished effect on the parameters examined. Even if data from our study are only preliminary indications, these findings indicate that the differences in PCOS phenotypes suggest that a different etiology of the syndrome is plausible and deserves to be further clarified. On this premise, new clinical approaches and therapeutic tools need to be investigated to treat women with PCOS with phenotype D since the therapeutical rationale, particularly for this PCOS sub-group, should be redefined.

## Figures and Tables

**Figure 1 biomedicines-11-01759-f001:**
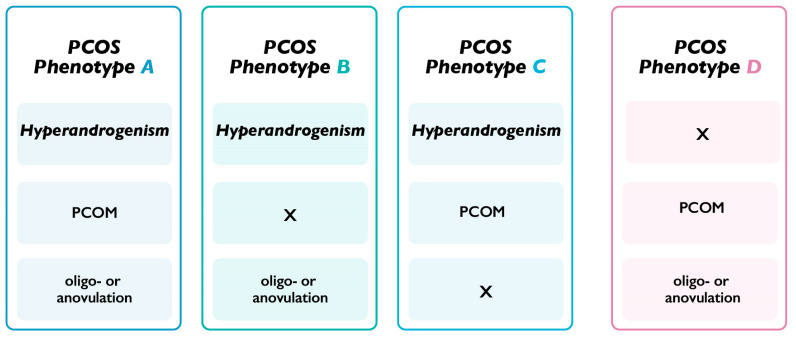
Rotterdam criteria for PCOS phenotypes. According to the Rotterdam consensus [2], polycystic ovarian syndrome (PCOS) is determined by the presence of at least two out of three of the following features: evidence of oligo- or anovulation, biochemical or clinical hyperandrogenism, and polycystic ovarian morphology (PCOM) (≥12 follicles measuring 2–9 mm in diameter and/or an ovarian volume >10 mL in at least one ovary).

**Table 1 biomedicines-11-01759-t001:** Baseline values.

	Hyperandrogenic PCOS (Phenotype A-B-C)	Non-Hyperandrogenic PCOS (Phenotype D)	*p*-Value
BMI (kg/m^2^)	28.3 (25.9–31.6)	24 (23–25)	0.00072
Endometrial thickness (mm)	3 (3–3)	8 (7–9)	<0.00001
Insulin (mIU/L)	18.5 (15.6–30)	12 (11–14)	<0.00001
Glucose (mg/dL)	89 (85.8–93.3)	79 (76.5–88)	0.01174
HOMA	4.1 (3.6–5.9)	2.5 (2.1–2.9)	<0.00001
FSH (mIU/mL)	5.2 (4.2–6.2)	4 (3–4.5)	0.00512
LH (mIU/mL)	11.1 (9.1–12.6)	3 (3–4.5)	<0.00001
LH/FSH	2.3 (1.8–2.5)	1 (0.8–1.2)	<0.00001
Testosterone (ng/dL)	92.5 (77.5–98.8)	34 (31.5–36.5)	<0.00001
SHBG (nmol/L)	44 (39–49.3)	88 (77–96)	<0.00001
Cholesterol (mg/dL)	198 (188.3–206.5)	155 (145.5–163.5)	0.00008
Triglycerides (mg/dL)	187.5 (130.3–247)	76 (66.5–86.5)	<0.00001

Table of the baseline values recorded for each study group before study start. Records indicated as median values, with the values of 25th and 75th percentile in parentheses. Statistical comparison of the baseline values of the two study groups is expressed with *p*-values. Body mass index (BMI), homeostasis model assessment (HOMA), follicle-stimulating hormone (FSH), luteinizing hormone (LH), sex-hormone-binding globulin (SHBG).

**Table 2 biomedicines-11-01759-t002:** Effect of myo-Ins treatment in both study groups.

	Hyperandrogenic PCOS (Phenotype A-B-C)			Non-Hyperandrogenic PCOS (Phenotype D)		
	T0	T6	*p*-Value	T0	T6	*p*-Value
BMI (kg/m^2^)	28.3 (25.9–31.6)	28 (26.6–31.7)	0.529	24 (23–25)	23 (23–25)	1
Endometrial thickness (mm)	3 (3–3)	6 (5.5–7)	**0.0004**	8 (7–9)	8 (6–8.5)	0.254
Insulin (mIU/L)	18.5 (15.6–30)	19.2 (14–20.9)	0.056	12 (11–14)	11 (10–12)	**0.009**
Glucose (mg/dL)	89 (85.8–93.3)	68 (64.5–76.3)	**0.0004**	79 (76.5–88)	78 (74.5–80)	0.099
HOMA	4.1 (3.6–5.9)	3 (2.5–3.6)	**0.0004**	2.5 (2.1–2.9)	2 (1.8–2.4)	**0.003**
FSH (mIU/mL)	5.2 (4.2–6.2)	6 (5.5–8.3)	0.031	4 (3–4.5)	3 (3–4)	0.084
LH (mIU/mL)	11.1 (9.1–12.6)	8.6 (7.9–9.4)	**0.001**	3 (3–4.5)	3 (2–4)	0.294
LH/FSH	2.3 (1.8–2.5)	1.4 (1.1–1.6)	**0.002**	1 (0.8–1.2)	1 (0.7–1.3)	0.569
Testosterone (ng/dL)	92.5 (77.5–98.8)	56.5 (45–62.8)	**0.001**	34 (31.5–36.5)	35 (32–37)	0.795
SHBG (nmol/L)	44 (39–49.3)	113 (94.3–123.3)	**0.0004**	88 (77–96)	100 (90–100)	**0.005**
Cholesterol (mg/dL)	198 (188.3–206.5)	175.5 (157–198)	0.147	155 (145.5–163.5)	156 (154–169.5)	0.134
Triglycerides (mg/dL)	187.5 (130.3–247)	149 (137.3–171)	**0.047**	76 (66.5–86.5)	95 (89.5–100)	**0.0003**

Table of the values recorded at baseline (T0) and after 6 months treatment with myo-inositol (myo-Ins) (T6) for each study group. Records indicated as median values, with the values of 25th and 75th percentile in parentheses. The *p*-values reported in the table refer to the statistical analysis of the variations measured after 6 months treatment in each study group. Body mass index (BMI), homeostasis model assessment (HOMA), follicle-stimulating hormone (FSH), luteinizing hormone (LH), sex-hormone-binding globulin (SHBG).

**Table 3 biomedicines-11-01759-t003:** Comparison of myo-Ins treatment in the study groups.

	Hyperandrogenic PCOS (Phenotype A-B-C)	Non-Hyperandrogenic PCOS (Phenotype D)	*p*-Value
BMI (kg/m^2^)	−0.2 (−0.8–0.1)	0 (0–0)	0.4009
Endometrial thickness (mm)	3 (2–4)	−1 (−2–0.5)	<0.00001
Insulin (mIU/L)	−3.4 (−9.0–1.3)	−1 (−4–0)	0.5157
Glucose (mg/dL)	−18.5 (−29–−12.3)	−6 (−11.5–3.0)	0.00048
HOMA	−1.2 (−3.1–−0.7)	−0.3 (−0.8–-0.1)	0.00758
FSH (mIU/mL)	1.1 (−0.2–3.6)	0 (−1.5–0.5)	0.007
LH (mIU/mL)	−1.7 (−3.5–−1)	0 (−2–1)	0.01778
LH/FSH	−0.9 (−1.4–−0.4)	0 (−0.6–0.6)	0.00244
Testosterone (ng/dL)	−33.5 (−45.3–−21.8)	1 (−3–3.5)	<0.00001
SHBG (nmol/L)	60.5 (57.5–79.5)	6 (1–23.5)	<0.00001
Cholesterol (mg/dL)	−23.5 (−46.8–11.3)	6 (0–12)	0.05486
Triglycerides (mg/dL)	−18 (−91.8–1.5)	22 (9.5–28.5)	0.00194

Table of the values referred to the variations recorded after treatment with myo-inositol (myo-Ins) for each study group. Records indicated as median values, with the values of 25th and 75th percentile in parentheses. The *p*-values reported in the table refer to the statistical analysis of the variations measured after 6 months treatment in each study group. Body mass index (BMI), homeostasis model assessment (HOMA), follicle-stimulating hormone (FSH), luteinizing hormone (LH), sex-hormone-binding globulin (SHBG).

## Data Availability

Data will be made available to the editors of the journal for review or query upon request.

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
