# Peer review of "Treatment with Myo-Inositol Does Not Improve the Clinical Features in All PCOS Phenotypes"

_biomedicines, 2023, doi:10.3390/biomedicines11061759_

Round 1
Reviewer 1 Report
This is an interesting retrospective study aimed to investigate the effects of myo-Ins treatment in different PCOS phenotypes. The hypothesis es well planned and the biological plausibility is explained, however, the data interpretation is not clear and I have son queries and suggestions:
1) In the PCOS Revised 2003 consensus on diagnostic criteria (Ref 2) the criteria used is “oligo-or anovulation” instead of “Menstrual cycle alteration”. If would suggest using the more recent and clear terminology used in the last ESHRE Guideline.
2) It is not clearly explained in the article how many patients are in each group of PCOS. The total sample appears to be very short. In my opinion, it is necessary to show the number of patients in each group and explain how the sample size was estimated, and know the power of the study and the probability of type I error
3) This is a retrospective non-randomized uncontrolled study, but in the second paragraph of the Discussion section it is stated that: “this is the first experimental trial comparing the effects of the same treatment with myo-Ins in different PCOS phenotypes”. I think this sentence should be changed
4) Why did the authors state that administration is ineffective in phenotype D if they detected changes in insulin, HOMA, and triglycerides?
5) Could the differences between both arms of the study at baseline have any influence on the results? How the authors controlled for those differences at baseline?
6) The sentence “ myo-Ins induced a significant increase of the endometrial thickness in a group of PCOS phenotypes A-B-C and a had a negligible effect in the group of PCOS phenotype D” should be also explained, especially when data of serum oestradiol levels are not available
7) The conclusion section appears as a summary instead of a real conclusion.
Author Response
This is an interesting retrospective study aimed to investigate the effects of myo-Ins treatment in different PCOS phenotypes. The hypothesis es well planned and the biological plausibility is explained, however, the data interpretation is not clear and I have son queries and suggestions:
1) In the PCOS Revised 2003 consensus on diagnostic criteria (Ref 2) the criteria used is “oligo-or anovulation” instead of “Menstrual cycle alteration”. If would suggest using the more recent and clear terminology used in the last ESHRE Guideline.
- Thank you for your comment. We edited the figure accordingly.
2) It is not clearly explained in the article how many patients are in each group of PCOS. The total sample appears to be very short. In my opinion, it is necessary to show the number of patients in each group and explain how the sample size was estimated, and know the power of the study and the probability of type I error.
- We specified the number of patients included in each study group. We also strongly agree that the number of total patients is limited so that we highlighted along the text that our results are just preliminary and further studies will be necessary to confirm the trend observed. Additionally, we did not calculate the sample size for the present study since no previous study investigating the effect of myo-Inositol on different PCOS phenotypes was already present in literature. We selected the patients from our database according to the inclusion/exclusion criteria trying to keep a proportion between the two study groups. The findings from this small sample of subject will be useful as basis to plan future double-blinded randomized controlled prospective trials on larger population. We specified this point both in methods and in conclusion sections.
3) This is a retrospective non-randomized uncontrolled study, but in the second paragraph of the Discussion section it is stated that: “this is the first experimental trial comparing the effects of the same treatment with myo-Ins in different PCOS phenotypes”. I think this sentence should be changed.
- Thank you for the comment. We changed the sentence according to your indication.
4) Why did the authors state that administration is ineffective in phenotype D if they detected changes in insulin, HOMA, and triglycerides?
- We changed the term “ineffective” and modified the two sentences of reference.
5) Could the differences between both arms of the study at baseline have any influence on the results? How the authors controlled for those differences at baseline?
- Thank you for the comment. We edited the material and methods indication of the baseline, and we added a sentence (lines 237-240) to better explain how the baseline characteristics were considered.
6) The sentence “myo-Ins induced a significant increase of the endometrial thickness in a group of PCOS phenotypes A-B-C and a had a negligible effect in the group of PCOS phenotype D” should be also explained, especially when data of serum estradiol levels are not available.
- We edited the paragraph accordingly. Considering that the estrogens levels were not analyzed in the study, we speculated on a possible explanation mechanism for the results described.
7) The conclusion section appears as a summary instead of a real conclusion.
- Thank you for your suggestion. We changed the conclusion section accordingly.
Reviewer 2 Report
General Comment: The manuscript submitted by Unfer et al aimed to investigate the effects produced by treatment with Myo-inositol in women with polycystic ovary syndrome phenotypes A-D. The rationale and hypothesis of this study is well described; however the data and conclusions are somewhat preliminary and not entirely convincing.
Comments:
1. the direct evidence and the detailed mechanisms for Myo-Inositol usage are not explained.
2. Mode of action of Myo-inositol needs to be discussed in more detail as it might affect different phenotypes.
3. PCOS phenotypes and their characteristics need to be explained in more detail for general readers.
4. Overall the manuscript is descriptive and premature and require more experimental evidences.
Author Response
General Comment: The manuscript submitted by Unfer et al aimed to investigate the effects produced by treatment with Myo-inositol in women with polycystic ovary syndrome phenotypes A-D. The rationale and hypothesis of this study is well described; however the data and conclusions are somewhat preliminary and not entirely convincing.
Comments:
- the direct evidence and the detailed mechanisms for Myo-Inositol usage are not explained.
- We elaborated upon the description of the methods accordingly.
- Mode of action of Myo-inositol needs to be discussed in more detail as it might affect different phenotypes.
-Thank you for your comment. We elaborated upon this aspect (lines 64-77).
- PCOS phenotypes and their characteristics need to be explained in more detail for general readers.
- We added a paragraph in the introduction (lines 40-46).
- Overall the manuscript is descriptive and premature and require more experimental evidence.
- Thank you for your suggestion. We agree that the population considered for the study is limited so that we highlighted along the text that our results are just preliminary and further studies will be necessary to confirm the trend observed.
Reviewer 3 Report
The introduction is based on contemporary publications, however, it should be extended and allow the reader to become familiar with the topic of research.
The methodology should be described more clearly.
The results are very limited.
The discussion is very long but contributes little important knowledge.
The conclusions are too general.
Author Response
The introduction is based on contemporary publications; however, it should be extended and allow the reader to become familiar with the topic of research.
- Thank you for the suggestion. We added a paragraph in the introduction (lines 40-46)
The methodology should be described more clearly.
- We elaborated upon the description of the methods accordingly.
The results are very limited.
- We edited the paragraph with more details on the results.
The discussion is very long but contributes little important knowledge.
- Thank you for your comment. We edited the discussion paragraph adding more information regarding the findings of the study.
The conclusions are too general.
- We changed the conclusion section accordingly.
Round 2
Reviewer 1 Report
The paper has now improved; however, I still find some problems:
1) Sentences like “In the present study, we demonstrate that 210 myo-Ins supplementation significantly reduced testosterone levels and increased SHBG 211 in H-PCOS thus confirming the proven efficacy of this treatment to counteract hyper-212 androgenism by improving hormonal parameters” is not supported for an observational study which only can find “association” and not “causality”. So that statement should be only supported by the results of a clinical trial.
2) Endometrial effects are also complicated to evaluate without serum estradiol levels and it should be explained in the study's weakness.
3) In this sense, I keep missing the last paragraph in the Discussion section with the strength and weaknesses of the study
4) Lastly, the Conclusion section is too large and the hypothesis should be not included in this section
Author Response
Dear referee, thanks for your comments and suggestions.
According with your queries:
Reviewer 1
1) Sentences like “In the present study, we demonstrate that 210 myo-Ins supplementation significantly reduced testosterone levels and increased SHBG 211 in H-PCOS thus confirming the proven efficacy of this treatment to counteract hyper-212 androgenism by improving hormonal parameters” is not supported for an observational study which only can find “association” and not “causality”. So that statement should be only supported by the results of a clinical trial.
- Thank you for your comment. We edited the paragraph accordingly (lines 205-209)
2) Endometrial effects are also complicated to evaluate without serum estradiol levels and it should be explained in the study's weakness.
3) In this sense, I keep missing the last paragraph in the Discussion section with the strength and weaknesses of the study.
- Thank you for the suggestion. We added a specific paragraph in the discussion (lines 326-333).
Please note that the measurement of estradiol levels is strongly controverse in the scientific community, considering that it represents a highly variable feature that significantly change in women depending on the day of the menstrual cycle in which the analysis is conducted. Additionally, we hypothesized a form of local hyperestrogenism that may not be extended at a systemic level and difficult to be revealed with serum analysis. Thus, the measurement of estradiol may not be as informative as expected.
4) Lastly, the Conclusion section is too large and the hypothesis should be not included in this section.
- We modified the section accordingly.
We hope that this second version of the manuscript could be suitable for publication on Biomedicines.
Thanks again for your revision.
Best regards.
Vittorio Unfer

Reviewer 2 Report
Thank you for addressing all the comments
Author Response
Thank you for your commitment in revising our manuscript.
We really appreciated.
Reviewer 3 Report
No further comments.
Author Response

(The authors gave the same response as above.)
